# “Mothers Should Have Freedom of Movement”—Citizens’ Attitudes Regarding Farrowing Housing Systems for Sows and Their Piglets

**DOI:** 10.3390/ani11123439

**Published:** 2021-12-02

**Authors:** Bianca Vandresen, Maria José Hötzel

**Affiliations:** Laboratório de Etologia Aplicada e Bem-Estar Animal, Universidade Federal de Santa Catarina, Rod. Admar Gonzaga 1346, Itacorubi, Florianópolis 88034-001, Brazil; vandresen.bianca@gmail.com

**Keywords:** animal welfare, farrowing crates, loose farrowing pens, outdoor, pigs, social license

## Abstract

**Simple Summary:**

Housing systems with increased freedom of movement for the sow have been proposed to replace farrowing crates. However, increased piglet mortality by crushing is the main barrier to the adoption of loose systems. Although animal welfare is a socially constructed concept, no study has investigated public opinion about alternative farrowing systems. We investigated Brazilian citizens’ attitudes towards farrowing crates, loose pens, and outdoor farrowing systems, and whether the increased risk of piglet mortality by crushing would influence the acceptability of loose pens to replace the crates. Attitudes towards loose farrowing were more negative than those towards outdoor housing, and more positive than those towards crates, with only a minor change when increased piglet mortality was considered. Participants had concerns about the sows’ restriction of movement in the crates, which they considered cruel and unnatural, and considered outdoor farrowing as the closest to the natural life of pigs. Importantly, loose pens did not address all participants’ concerns about farrowing housing systems, especially socialization between sows and expression of maternal behaviors. Our findings indicate that the development of alternative farrowing housing systems is necessary, but it has to be in line with the public’s expectations to be socially sustainable.

**Abstract:**

Farrowing crates are the most common housing for lactating sows throughout the world, despite known public opposition to housing systems that deprive animals’ freedom to move. This study aimed to investigate Brazilian citizens’ attitudes towards three farrowing housing systems (crates, loose pens, and outdoors). Data were obtained via an online questionnaire containing open- and close-ended questions. Participants’ (*n* = 1171) attitudes were most negative towards the crates and most positive towards outdoor farrowing, and positively correlated with perceived sows’ quality of life. Participants overwhelmingly rejected the use of farrowing crates, and most supported a proposal to move from farrowing crates to loose pens, even when informed that this entailed an increased risk of piglets’ mortality. Participants’ views were underpinned by concerns about sows’ freedom of movement, behavioral freedom and naturalness, and the belief that it is possible to develop and manage housing that prevents piglet crushing that does not involve confining the sow. Furthermore, loose farrowing pens may not fully address all concerns expressed by participants regarding farrowing housing, which included the possibility of allowing sows to socialize and express maternal behaviors. We conclude that maintaining farrowing crates may erode the pig industry’s social license.

## 1. Introduction

Gestation and farrowing crates confine the sow in a way that she can stand up and lie down but cannot turn around or walk. Gestation crates are used in intensive pig production systems to house sows during gestation (approximately 114 days), whereas farrowing crates are used to house sows from around a week before farrowing until piglets are weaned (most usually imposed between 21 and 35 days after birth). In the farrowing crate, the piglets can circulate around the sow’s crate in a pen of around 4 m^2^. The implications of confinement housing for animal welfare and food production have been a growing matter of debate among researchers, citizens, and policymakers over the years. For example, public rejection of farm animal confinement may influence consumers to pay more for products from systems that provide a higher level of animal welfare [1], or result in private regulations and animal welfare auditing programs to their suppliers [2], or voting behavior in support of legislation that protects animals [3]. In the case of gestation crates, several countries and some of the largest food companies in the world have banned their use [4]. In contrast, the farrowing crate is still the most common housing for lactating sows (Brazil [5]; European Union [6]; New Zealand [7]; United States of America [8]), except in Sweden, Switzerland, and Norway, the only countries where the use of farrowing crates is currently prohibited. The lack of urgency to change the status quo of farrowing crates may be associated with the lack of research on the societal opinion on that matter. The few studies that specifically investigated public opinion regarding the use of farrowing crates showed public opposition (Germany [9]; Brazil [10]).

More recently, political debate about the use of farrowing crates emerged in Austria [11], Germany [12], New Zealand [13], and the United Kingdom [14], which are currently discussing or have recently established regulations to phase out farrowing crates. The European Parliament is also discussing a ban on the use of farrowing crates by 2027 [15], which is supported by citizens’ initiatives like “End the Cage Age” (www.endthecageage.eu, accessed on 4 November 2021). However, many countries still have not started this discussion. For instance, in Brazil, one of the largest pork producers and exporters in the world [16], the federal government recently published the Normative Ruling n. 113 [17] establishing several management changes aiming at improving pig welfare, including the replacement of gestation crates with group gestation by 2045, but this same document states that the use of farrowing crates will continue to be permitted. Similarly, regulations that prohibit the use of cages to house some farm animals (e.g., laying hens, veal calves, and gestation sows) allow the use of farrowing crates in the United States [18].

Alternatives to the farrowing crate are outdoor farrowing and loose farrowing pens. Outdoor farrowing allows the sow to walk freely in paddocks and perform highly motivated behaviors including nesting, walking, rooting, and grazing [19]. In the United Kingdom, although the use of farrowing crates is still allowed, almost half of the sows on commercial farms are housed in outdoor farrowing systems [20]. Loose farrowing pens are the main alternative to replace the farrowing crates in confinement farrowing systems. This housing offers some space for the sow to move and interact with the piglets [21], and some models also allow nesting behavior.

Piglet mortality by crushing is an important part of the farrowing housing debate, because the proponents of the farrowing crates argue that the crates are necessary to prevent piglet crushing and that the change to alternative systems would result in the death of many piglets, which would be economically unviable for pig producers and detrimental to piglets’ welfare [22,23]. Piglet crushing is considered a main cause of piglet preweaning mortality; it occurs when the sow lies over her piglets [24], which is most common in the early days after farrowing [25]. However, some studies have questioned the efficiency and need of farrowing crates to prevent piglet crushing, since it has been shown that neonatal piglet mortality is associated with several factors besides housing, namely farrowing management, birth weight, and litter size [26,27].

Alternative housing systems recommended to solve animal welfare problems need to address the social concerns originating from the demand for change in the farrowing crate system, otherwise they may prove to be socially unsustainable and threaten the animal industries’ social license to operate in the long term [4,28]. In the case of farrowing housing systems for pigs, it is not known to what extent the loose pens accommodate public concerns regarding sows’ and piglets’ welfare. This study aimed to investigate Brazilian citizens’ attitudes regarding farrowing housing for sows and the influence of providing information about the risk of piglet mortality by crushing in loose farrowing pens on the acceptability of this system.

## 2. Materials and Methods

Research involving humans in Brazil is regulated by Regulations n. 466/2012 and 510/2016 from the Brazilian National Health Council; the later states that surveys on public opinion that do not identify participants are not evaluated by the CEP/CONEP (Comissão Nacional de Ética em Pesquisa) system. All procedures followed the ethical principles established by this regulation. Before answering the questionnaire, participants had to read an informed consent statement and accept the conditions of the study, i.e., that participants were not identified, participation in the survey was voluntary and anonymous, that data would be used exclusively for scientific research, treated confidentially, and that they had the opportunity to withdraw at any moment by not sending the form. All participants were required to give consent about their participation by clicking a button saying “I agree to participate in the research” before taking the survey.

### 2.1. Participants’ Recruitment

The online survey was in Brazilian Portuguese and conducted using the Google forms platform. An advertisement saying “Collaborate with our research on animals. Access the link to participate” was shared during the months of April and May 2020 in an Instagram account created exclusively to advertise the questionnaire and without further information on the survey subject. By the end of participants’ recruitment, we strategically targeted participants to match the age and sex distribution of the Brazilian population [29]. Conditions to participate in the research were to be a Brazilian citizen, at least 18 years old, and not be totally against rearing animals to produce food.

### 2.2. Survey Methodology

The questionnaire started by asking participants about their sociodemographic information: sex (male or female), age (18–25, 26–35, 36–45, 46–55, 56–65, or over 65 years old), monthly income, region (North, Northeast, South, Southeast, or Midwest), whether they lived in a rural or urban area, education (primary school, high school, technical course, undergraduate degree, graduate degree), their involvement with livestock production (no involvement, grew up in an agricultural environment, involved as a farmer, involved as a professional, involved as a student) and whether they currently had, had in the past, or had never had a dog, cat, or pig.

Next, participants were then invited to read a short text about farrowing housing (crates, loose pens, and outdoors) and were provided six images displayed on graphics interchange format (GIF) to illustrate each housing system. The questionnaire with the text and images of the farrowing housing systems was translated to English and provided as Appendix A. After reading the text, each participant was randomly assigned to answer about one of the farrowing systems described: farrowing crates, loose farrowing pens, or outdoor farrowing. Participants were asked to indicate whether they knew about the farrowing system previously to participating in the survey, and to rate on 5-point Likert scales: “Do you consider this farrowing system appropriate?” (1 = totally inappropriate; 5 = totally appropriate), “Do you approve this farrowing system?” (1 = totally disapprove, 5 = totally approve), and “Do you consider this farrowing system acceptable?” (1 = totally unacceptable; 5 = totally acceptable). This was followed by an open question asking participants to justify their answer to these questions. Participants were also asked to rate how they evaluated the quality of life of the sow and piglets in the farrowing system on two 5-point Likert scales (1 = very low; 5 = very high).

The following section presented a hypothetical scenario to all participants, “A medium-sized company that uses farrowing crates will increase its number of sows. The company intends to adopt loose farrowing pens for these animals.” One-third of participants were provided with no extra information; one-third were informed that “In a preliminary test of the loose farrowing pens, the company identified an increase in mortality of piglets by crushing, from 10% to 12.5%”; and the other third had the same information but with an increase in mortality of piglets by crushing from 10% to 15%. After the text, all participants were asked to choose one of the two options, “Yes, I believe the company should change to the loose farrowing system” or “No, I believe the company should keep using the farrowing crates”. Participants were then asked to justify their answer in an open question. The increases in piglet mortality by crushing proposed in the scenarios were based on studies that compared piglet crushing in farrowing crates and loose farrowing pens, e.g., [30,31].

Before the next section, participants had to read some statements about farrowing housing systems and indicate whether they we true or false based on the text previously provided in the questionnaire. In this question, participants also had to pass an attention check: “Choose the option ‘false’ to validate your answers”. Participants were also asked their opinion about the use of animals for food production, and consistency of the answers with the statements, “I am totally opposed to rearing animals to produce food”, “I support rearing animals to produce food without restrictions” or “I support rearing animals to produce food, provided it is performed in an ethical manner”. Participants totally opposed to rearing animals to produce food were excluded from the analysis.

In the last section, participants were asked about their meat consumption preferences: whether they eat pork (yes or no), how many days a week they eat meat (pork, beef, chicken, or fish) (none, rarely, 1 to 2 days, 3 to 4 days, 5 to 7 days), how frequently they think about how animals are raised when they eat meat (on a 5-point Likert scale; 1 = never; 5 = always), the importance of eating meat for them (on a 5-point Likert scale; 1 = not important; 5 = very important), and how much they believed that Brazilian citizens would be willing to pay more for pork from loose farrowing pens and outdoor farrowing compared to the current price (5% more, 10% more, or 30% more). Participants also answered from which farrowing housing system previously described they would rather buy pork if there was no difference in price; they were asked to justify their answer in an open question.

### 2.3. Data Analysis

As the closed questions were mandatory, participants had to answer all questions to complete and send the questionnaire. Participants were excluded from the analysis if they were totally against rearing animals to produce food (*n* = 183) or if they failed an attention check (*n* = 63), resulting in a sample of 1171 participants.

Some of participants’ demographic characteristics were grouped for the data analysis. Participants’ education information was grouped in two categories: up to high school (primary school, high school) and post-secondary education (technical course, undergraduate degree, graduate degree). Participants’ involvement with livestock production was grouped in three categories: no involvement, grew up in an agricultural environment, and involved as a farmer, professional, or student. Participants who currently had, had in the past, or had never had a dog and/or a cat were grouped as participants who currently had, had in the past, or had never had a pet.

Cronbach’s alpha was used to assess consistency of the answers to “Do you consider this farrowing system appropriate?”, “Do you approve this farrowing system?”, and “Do you consider this farrowing system acceptable?”; as the alpha coefficient was 0.98, the average of the three answers created a mean for each participant and was treated as their attitude score towards the farrowing system.

The effects of each of the three treatments (crates, loose pens, and outdoors), participants’ sociodemographic characteristics (age, sex, education, region, urban/rural, and income), involvement with livestock production; whether they currently had, had in the past, or had never had a pet or a pig; meat consumption behavior (days per week that they ate meat), awareness of the farrowing system previous to participating in the survey, and their interactions on attitude were tested using ANOVA. Least-square means and standard errors are presented in the results, with significance declared for *p* < 0.01. Associations between participants’ sociodemographic characteristics (age, sex, education, region, urban/rural, and income), involvement with livestock production, meat consumption behavior (days per week that they ate meat, whether they ate pork), information on piglet crushing (no information, increase of 2.5%, or increase of 5%) and support for the pig company’s decision (move to loose farrowing pens or keep using farrowing crates) were tested using binary logistic regressions. Participants’ evaluation of sows’ and piglets’ quality of life were compared using the Wilcoxon test. All statistical analyses were performed with the R software (R Development CoreTeam, Vienna, Austria, 2011).

The answers to the three open questions (justification of attitude towards the farrowing system; justification of support for the pig company to move to loose farrowing pens or keep using farrowing crates; and justification of which system they would prefer to buy pork from) were submitted to inductive thematic analysis, which involves careful reading and rereading of the text to capture complexities of meanings in the data [32]. Both authors familiarized themselves with the data and individually built the initial coding of all answers. Then the authors compared their results and discussed any discrepancies and ambiguities until agreement was reached. Both authors worked together on identifying emerging patterns across the data to develop themes using a semantic approach. Responses from 1134 participants were used as not all participants answered the open questions or provided a meaningful answer. Extracts of participants’ answers considered representative of the themes are provided in the results sections alongside participants’ ID (e.g., P50). The quotes were translated from Brazilian Portuguese to English by BV and reviewed by MJH.

## 3. Results

Participants’ demographic data (Table 1) of the total sample corresponded to the Brazilian population [29] except for education, and was accordingly distributed within the treatment groups: participants who answered about farrowing crates (FC), loose farrowing pens (LP), or outdoor farrowing (OF). Most participants were not involved with livestock production (68% of total participants; 74% of FC; 64% of LP; 67% of OF), some grew up in an agricultural environment (19% of total participants; 17% of FC; 19% of LP; 20% of OF), and others identified as farmers, professionals, or students currently involved with livestock production (13% of total participants; 9% of FC; 17% of LP; 13% of OF). Most participants (86%) consumed pork and around half (54%) considered eating meat important or very important (neutral = 27%; not much/not important = 19%). Few participants never (3%) or rarely (13%) consumed meat (pork, beef, chicken, or fish), and other participants consumed meat at least one day a week (1 to 2 days = 18%; 3 to 4 days = 20%; 5 to 7 days = 46%). Fifty percent of the participants answered that they always or often thought about how animals were raised when they ate meat (sometimes = 25%; a few times or never = 25%).

Attitude scores were most negative toward the farrowing crates and most positive toward the outdoor farrowing system (Figure 1). Housing system (F2, 1160 = 640.5, *p* < 0.001), participant’s sex (F1, 1160 = 38.2, *p* < 0.001), and days per week they ate meat (F4, 1160 = 4.35 *p* < 0.01) influenced participants’ attitude scores. There was also an interaction between sex and farrowing system (F1, 1160 = 11.9, *p* < 0.001), with women showing lower attitude scores than men on the farrowing crate and loose farrowing pen systems (Tukey’s HSD test, *p* < 0.0001) but not on the outdoor system. Previous awareness of the housing system also influenced attitude (F1, 1160 = 8.7, *p* < 0.003); participants that had no previous awareness of the housing system had more negative attitude scores than participants that knew about the system (Table 2).

Participants’ attitudes towards a given farrowing system and assessment of the quality of life of sows in the same system were positively correlated: farrowing crates (Spearman r = 0.68, S = 33,329, *p* < 0.001), loose farrowing pens (Spearman r = 0.45, S = 51,490, *p* < 0.001), and outdoor farrowing (Spearman r = 0.53, S = 46,999, *p* < 0.001). Participants considered the sow’s quality of life worse than piglets’ in the farrowing crates (V = 273, *p* < 0.001), but not in the loose farrowing pens (V = 2332.5, *p* = 0.071), or in the outdoor farrowing (V = 1197, *p* = 0.53).

The greatest support for the proposed scenario of a company’s move to loose farrowing pens instead of maintaining the use of farrowing crates was among the group that received no information about piglet crushing (90% participants), followed by 86% that were informed that the move would result in a 5% increase in piglet mortality, and 82% participants that were informed that the move would result in a 2.5% increase in piglet mortality. Logistic regression indicated that participants were less likely to support the company’s move to the loose farrowing pens if they were informed that piglet crushing would increase by 2.5% (OR = 0.92, 97.5% CI = 0.88–0.97, *p* < 0.002) and if they considered meat eating very important (OR = 0.90, 97.5% CI = 0.86–0.95, *p* < 0.002).

Most participants (87%) preferred to buy pork from the outdoor farrowing system, 10% from the loose farrowing system, and 3% from the farrowing crate system, if there was no difference in the price of pork. Most participants believed that, compared to the current price of pork, Brazilian citizens would be willing to pay more for pork from loose farrowing pens (5% more = 40% participants, 10% more = 25% participants, and 30% = 4% participants) and outdoor farrowing (5% more = 28% participants, 10% more = 38% participants, and 30% = 12% participants).

Thematic analysis of participants’ open answers yielded four key themes (Figure 2): views regarding animal welfare in farrowing systems; piglet mortality and sows’ welfare; participants’ role as agents of change in animal production systems; reasons to keep sows in crates. The themes are presented below with illustrative extracts of the data.

### 3.1. Views Regarding Animal Welfare in Farrowing Systems

Participants’ views about farrowing systems focused on ethical concerns about animals’ freedom. Participants also discussed the implications of farrowing systems over all three aspects of animal welfare: biological functioning, affective states, and naturalness, and pointed to associations between those aspects.

Participants described farrowing crates as ethically wrong due to the negative impact on animals’ quality of life and freedom of movement. Many used the words “cruel”, “inhumane”, and “suffering” to describe farrowing crates (“*Mothers should have freedom of movement. Space to move around and feed their offspring. This system is cruel*.”—p123). The alternative farrowing systems (loose farrowing pens and outdoor farrowing) were described as ethically superior. Participants described these systems as “more ethical”, “fair”, and “decent”. Some participants associated alternative farrowing systems with sows’ positive feelings (e.g., happiness: “*I believe that free animals are healthier and happier.*”—p473) and considered that animals should be happy (“*If the animal has the chance to be happy while it is alive, it is very comforting, as well as guaranteeing healthier meat. Just like humans, the most important thing is not how long we live, but how we live.*”—p931). In contrast, farrowing crates were related to sows’ negative feelings (e.g., depression: “*The animal needs freedom of movement, because she can also become depressed.*”—p375).

Restriction of space, especially regarding the sow, was a main concern about farrowing crates (“*The confinement system and management of the sows in this way benefits the piglets but causes a lot of stress to the sows because they are in this confined management*.”—p143). Participants considered that freedom of movement is a basic need (some used the term “right”), which should be guaranteed for all living beings (“*All living beings have the right to be free.*”—p673; “*The animal is a living being that must be treated accordingly. In this crate they are treated like robots that must fulfil their role, with no right to a natural environment.*”—p108). Allowing animals to perform natural behaviors and social interactions was described as respecting their dignity, which participants considered impossible in farrowing crates due to the restriction of space (“*The mother sow cannot even change positions, which is likely to cause a lot of pain, and she has no proper contact with her offspring.*”—p121). Provision of space was mentioned by participants who had positive and negative attitudes to the loose farrowing pens (“*Both systems [farrowing crates and loose pens] have very little space, but between them, the loose farrowing pen is better because it has more space.*”—p410; “*The [loose farrowing] system is still very restrictive to the animal.*”—p576; “*I still think it’s too little space.*”—p544).

Support for the loose farrowing system was often contrasted with negative attitudes towards farrowing crates (“*With the loose farrowing system, the animal is limited. Spending your whole life in a space of 6 to 9 [square meters] is cruel. And even though it is not ideal, having a little more mobility is better than being limited, without being able to move.*”—p515; “*The production of the farrowing crate should be forbidden, the loose one is bad enough, the farrowing crate should be banned*”—p606). Some participants stated that although the loose farrowing housing provided more space for the sow than the crates, they still had some concerns about it (“*It is a prison, regardless of having a little room for the sow to move around.*”—p645). Some participants explicitly stated that the loose farrowing system was better than the farrowing crates, but not good per se (“*Better loose [pens] than crate, but still awful.*”—p497; “*A minimum of freedom for the animal, but is not ideal.”*—p428). Others thought that the loose farrowing pen was no improvement at all compared to the farrowing crates (*“(…) I don’t know if the system of loose farrowing is better than the other [farrowing crates]; I see it as equally perverse: it is a choice between the life of the offspring or the mother. Neither system should even be considered.*”—p278).

Some participants considered outdoor farrowing the best alternative (“*The loose farrowing pen is already better than the crate, because the sow is not completely immobile, but the ideal would be the outdoor housing, so the two types mentioned above are totally unacceptable.*”—p478). Likewise, when considering the proposal of the company’s adoption of the loose farrowing pens, some participants considered that it should move to outdoor farrowing instead (“*I believe that companies should adopt the outdoor farrowing system once and for all… Why keep them in a smaller place (even if larger than the crate), when there is an outdoor option?*”—p529). Participants praised the naturalness of outdoor access (“*Outdoors respects the animal’s integrity, does not generate fear and will not cause traumas. It resembles the natural life of the animal.*”—p1112) and associated it with healthier diets and physical activity, which they believed would result in healthier animals and better meat quality (“*The animals, being free, are happier… I don’t understand much about it, but I believe that happy animals must be healthier, providing a better quality of meat.*”—p1106).

### 3.2. Piglet Mortality and Sows’ Welfare

Participants argued that preventing piglet crushing did not justify keeping the sows in crates, offering multiple reasons, including sow welfare *(“(…) The economic loss of the farmer cannot be a reason to subject the sow to a life of mistreatment.*”—p1162; “*Piglet crushing is a fatality that we cannot control, but constantly inseminating the same sow and keeping her locked up for most of her life is something that depends exclusively on us humans and therefore I believe we could avoid this suffering.*”—p663), views on piglet crushing (“*Crushing, it seems, is a natural thing to happen and cannot be used to justify the adoption of a model as cruel as the crate.*”—p707; “*A small percentage of offspring deaths should not be used as a standard for maintaining exclusive confinement rearing*.”—p357) and the belief that there are solutions to piglet crushing other than confining the sow (“*Piglet mortality cannot be the basis for leaving the sows in these horrible conditions. Other alternatives for lower piglet mortality must be found.*”—p167).

One main argument used by participants was that the sow should be free to move even if this resulted in a higher risk of piglets being crushed (“*Unfortunately there is the risk [of piglet crushing], thinking of the mother, who spends time locked up, immobile, nursing. I prefer to see them released even with some occasional deaths.*”—p906); some explicitly described the sow’s freedom as a fair price to pay for the welfare of the sow and piglets that remained (“*Better welfare of the mother, even if it costs the lives of a few piglets.*”—p*945; “It is a low percentage, compared to the quality of life of the animals.*”—p1048). Some participants considered that the increase of piglet mortality is a cost worth paying to move to alternative systems, since it also occurs in farrowing crates (“*It’s a 2.5% increase, but the number is still very close to 10%, the losses won’t be that much bigger, but the sow’s quality of life will be 500% higher*.”—p1022). Additionally, some participants considered piglets’ crushing a natural event and therefore acceptable in a production system (“*I believe that this [loose farrowing pens] is a better system than the crates because it would reduce the stress on the animal and she could at least walk, and the death of piglets is something to be accepted, being a common thing in nature. Just like rain destroys a crop.*”—p572).

Participants also conjectured that piglet crushing was a consequence of the loss of natural maternal behavior due to the selection of sows in the traditional farrowing system (“*The sows are probably no longer used to being loose with their piglets, thus causing their death by crushing. They have been raising animals confined like this for years, something must have changed in them so that they are no longer able to care for their offspring, since they have always been confined in tiny places where they could barely move.*”—p1032). In the same line of thought, some assumed that sows would adapt to alternative farrowing systems and piglets crushing would decrease with time (“*The sows will get used to the new type of rearing and the crushing losses will decrease*.”—p957).

The proportion of piglets crushed was not clear to all participants; some said that they did not know how many crushed piglets could be considered too many, given their lack of knowledge about the issue (“*Actually I’m not sure how many 10% to 15% of dead piglets represents, if you could make a considerable profit with the survivors.*”—p321).

Participants believed that stakeholders of the pig production chain are responsible for developing solutions to prevent piglets’ crushing in alternative farrowing systems. Participants cited farmers (“*I think it is essential that producers worry more and more about animal welfare. (…)*”—p967), industry (“*If the company is large and is going to increase production, it doesn’t make sense to increase the loss rate. It is more worthwhile for it to look for ways around the problem to change the system.*”—p1155) and scientists (“*New investigations could be made to reduce to mortality, which is natural if the environment is not well controlled.*”—p1005). Some participants expressed the same ideas with optimistic tone, implying that these stakeholders would be able to make these changes (“*It is possible that the continued practice will identify opportunities for improvement that can result in a reduction of the crushing rate.*”—p923; “*The company can move to the loose farrowing system and from the start try to find a way to avoid the death of piglets.*”—p1115), yet others seemed more accusatory (“*It is certainly possible to develop a system that avoids loss of piglets by crushing. All you need is to be creative and find an alternative way to solve the issue, one that does not require confining the sow.*”—p1101; “*There are other ways to avoid piglet mortality. Working on it, it is possible to arrive at reasonable solutions.*”—p374) and some argued that the real reason for the use of restrictive housing is to increase production and not the welfare of piglets *(“(…) The justification of the crate is obviously for profit, not for the welfare of the piglets, which will be slaughtered.*”—p638; “*Causing any kind of animal suffering for profit is immoral, unethical, inhumane, and unacceptable.*”—p88).

### 3.3. Participants’ Role as Agents of Change in Animal Production Systems

Some participants recognized their role as consumers to support animal products and producers that respect and promote animal welfare *(“(…) If I know where the products I consume come from, I can contribute to companies that act in a way that I believe make the world more ethical, sustainable, responsible, and fair.*”—p998). Some participants emphasized the moral obligation of ensuring a good quality of life of animals that are used to produce food for humans (“*I think it is a completely unacceptable and unfair system for animals, which are living beings like us and deserve at least respect, since they end up serving as a product for human beings against their will, not to mention that no living being deserves this kind of life…*”—p75). Participants also considered that farmed pigs have a short lifespan and therefore deserve to have a good quality of life (“*[…] They give birth for the benefit of the producer, they deserve at least in the short period of their life, a decent life.*”—p374; “*Even if they are bred to be slaughtered, they deserve quality in the short time they have to live.*”—p987). Some participants said they felt guilty about eating meat, with some arguing that promoting animal welfare would make them feel less guilty (“*Despite the hypocrisy of continuing to eat meat, I believe that knowing that the animal had some sense of contact with nature would ease my conscience*.”—p186; “*I think I would eat it without so much guilt.*”—p894).

Some stated that they would be willing to pay more for pork from non-crated farrowing systems and compared it with other animal food products (“*I already consume eggs coming only from free-range chickens, I would do the same with pork. And if it was much more expensive, I would reduce my total meat consumption to purchase this product.*”—p322; “*I think that the pig has the right to be able to sunbathe, dig holes, graze, and have other habits typical of the species. For all species like chickens, goats, sheep, or cattle, I would pay extra for the animal to have a decent life without unnecessary suffering.*”—p121). Participants also stated that they would feel better consuming meat of an animal that had a good life (“*To know that the animal had a healthy life and was closer to a free life comforts me.*”—p979), which would influence their choice when buying pork (“*(…) But in the case of choosing which animal feeds my body, I prefer to know that it has been free and peaceful during its life.*”—p315).

Participants associated animal welfare with meat quality, which was another reason to prefer pork from alternative farrowing systems (“*Besides the quality of life of the animals, I believe that the system is favorable to a better quality of the meat, since stressed and mistreated animals have a more contaminated meat due to anger, fear, and mistreatment.*”—p1052; “*Better quality of life for the animal that is reflected in the meat.*”—p829). Some participants considered the better quality of meat as a reason to accept the increase of piglets’ crushing (“*Although there is more death by crushing, I still believe that this system is the best, because it has a better quality of meat.*”—p671).

Some considered the change to alternative farrowing systems as an opportunity for producers to benefit from a better reputation of their products (“*You could even use it as marketing with a cool catch phrase. ‘We are beyond meat’, ‘We are more than production’, something like that. You could even charge more for rearing method*.”—p175). Others considered this change as necessary and inevitable for producers because of social concerns about animal welfare (“*More freedom for the animals, animal welfare is the future for these producers, either they adapt or they will be made accountable later.*”—p122; “*Today we are changing to vegan because we can’t stand to see so much suffering… the company that gives better living conditions has more market.*”—p633).

### 3.4. Reasons to Keep Sows in Crates

Some participants considered the welfare of the piglets that could be crushed (“*With the existent risk of piglet death, I believe that the farrowing crate is better for the welfare of both the piglets and the mother, even if the mother’s freedom is limited.*”—p103), and others were concerned about the economic impacts of losing piglets (“*The higher the mortality of piglets, the less profit. You are not in business to not make profit. Increased piglet mortality generates losses. It’s beautiful in theory, but it doesn’t work in practice. The Brazilian consumer is not concerned with the quality of life of the animal, but with how much they pay for the final product. If this meat is more expensive, just like free range eggs, consumers will prefer the cheaper product.*”—p708).

Participants who defended the use of farrowing crates also based their arguments on the risks of changing the traditional system to an alternative one, mostly because of the consequences for productivity (“*Maybe, given the demand for food, this kind of housing is necessary. I am very practical!*”—p34; “*It enables higher productivity per square meter.*”—p99), animal management (“*The more confined the animal is, the greater sanitary control will be carried out.*”—p570), and biosecurity (“*This system [outdoor farrowing] is only valid for those who live on farms and ranches, otherwise it is not acceptable because of the dirtiness*.”—p1149). Some participants also thought that it could be difficult for farmers to change the farrowing system (“*The high cost prevents small farmers from using this method!*”—p139), a thought shared by some participants who supported the change to alternative systems (“*For the farmer it may be more difficult/laborious or some loss may occur. But for the animal it is better.*”—p954; “*We have to respect nature and the closest thing to it is the outdoor farrowing system, not only farrowing but all animal husbandry should be outdoors. Now the question is: is there enough space to meet the demands of this system?*”—p1089).

## 4. Discussion

The assessment of the different farrowing housing systems was overwhelmingly based on concern about the welfare of the sows, which participants rated as most negative in farrowing crates and most positive in the outdoor housing. The preference for the loose pens over the farrowing crates was maintained even when participants were faced with the dilemma that providing more space for sows could incur some piglet mortality by crushing. These findings corroborate and expand previous findings [9,10], indicating that farrowing crates do not have societal support and that the use of piglet mortality cannot be used as justification to maintain this system.

Framing theory is based on the premise that an issue can be viewed from multiple perspectives [34]; issue framing has been explored as a strategic tool in political narratives [35,36]. The mainstream political narrative of farrowing housing systems is grounded on the argument that preventing piglet mortality by crushing is a priority and farrowing crates are the best housing system to achieve this while guaranteeing the sows’ and piglets’ welfare [23]. As described by Canadian pig farmers, farrowing crates are “one of the greatest animal welfare tools that exist” [37]. Our findings indicate that the pig industry needs to pay attention to the overwhelming public rejection of housing that limits animals’ freedom to move, socialize, and perform other natural behaviors, shown in this and countless other studies, e.g., [9,10,38,39], and the public’s perspective on the piglet crushing issue shown here. The effort made by the pig industry to maintain its social license by transitioning to group gestation housing entailed large financial, technical, political, and social investment, but the gains in terms of social support may be undermined by the reticence to move away from farrowing crates.

Public opposition to housing that prevents animals from moving freely and the clear preference for outdoor systems were shown in this and other studies, e.g., [38,40]; this calls for reflection on the steps to be taken to replace farrowing crates. Participants’ concern for the welfare of the sow indicates that moving towards sow-friendly housing systems would be better suited to societal expectations about farrowing housing systems. In line with our results, studies have identified that improving sow welfare in loose farrowing systems has positive effects on piglet welfare and growth rates [41]. In particular, our findings suggest that consumers may not support systems that use temporary crating, i.e., the confinement of sows in crates during parturition and early postpartum. The purpose of temporary crating is to decrease piglet mortality in loose farrowing systems [42]. A similar practice is used in some group gestation housing systems, which confine the sows in crates following insemination and during early gestation to preserve embryo survival [38]. Farrowing crates deprive the sows of fulfilling the motivation to perform maternal behaviors such as nesting and care of the newborn offspring, which not only causes stress [43] but also deprives sows of some of the few opportunities they have to experience positive emotions in commercial farms. Consumers value outdoor systems in part because they see animals as “happy” in these systems (shown in this survey and previously [40,44]). Philosophers and scientists increasingly discuss the importance of positive emotions in the context of farm animal welfare and try to devise ways to incorporate environmental features that allow positive emotions in intensive livestock production systems [45,46]. Looking into the future, the transition away from crate systems must seek to incorporate aspects that add positive emotions, rather than simply focusing on avoiding suffering. Half-way solutions may cost a great amount in time and financial investment with questionable returns to farmers, given that our findings and others [47,48] indicate that they may not settle the issue for consumers. In contrast, stakeholders should consider investing in systems that allow socialization between sows and the litter [21], among sows in get-away systems, and between litters in multi-suckling systems [49], which incorporate some aspects pointed to as relevant for participants in this study. Future studies on public opinion should investigate these and other aspects of sows’ farrowing housing to help guide the pig industry during the transition away from farrowing crates to more socially sustainable systems.

Some participants in this survey considered some piglet mortality acceptable in the context of transitioning to cage-free systems but expected it to decrease over time with the maturity of the farrowing system. Indeed, the varying results regarding piglet crushing in farrowing systems [25,31] may reflect the variation in the degree of maturity of cage-free farrowing systems in different countries and studies, and may suggest that farmers’ experience with loose systems may indeed improve piglet survival. Schuck-Paim et al. [50], for example, showed that the cumulative mortality in cage-free aviaries decreased over the years of experience with the housing system, while mortality in caged systems did not change, resulting in no differences in mortality between caged and cage-free aviaries in recent years. Importantly, multiple aspects of farrowing systems other than housing are risk factors that can be targeted to reduce piglet mortality. For example, it has been shown that many aspects of farrowing management influence piglet survival [51,52,53]. Also to be considered are large litters and the associated variation in piglet birth weight and the greater risk of piglet mortality in smaller piglets [54]. The pig industry has advocated for systematic genetic selection for hyperprolificity to increase the number of piglets weaned per sow, despite awareness of the risk of increased piglet mortality in large litters [55], indicating that individual piglet survival has been a relatively low priority in the production context.

Our finding suggests that information about how the meat has been produced may influence attitudes towards meat consumption, given that participants stated that they would feel less guilty by consuming pork from alternative farrowing systems. The conflict of being concerned about animal welfare and consuming meat is addressed as the meat paradox [56]. Cognitive dissonance is the attempt to reduce the deviance between beliefs and behaviors [57]. The belief–behavior inconsistency related to meat-eating is based on the concept of meat-related cognitive dissonance (MRCD), and the strategies used to prevent or reduce MRCD are reviewed by Rothgerber and Rosenfeld [58]. Briefly, people may try to reduce their guilt about consuming meat in multiple ways, for example, by reducing the amount of meat in their diet [59] or decreasing their belief in the capacity for suffering of animals categorized as food [60]. Some participants in this and other studies [61,62] showed a preference for products they perceived to promote higher animal welfare. Some elements that lead to the perception of higher pig welfare are freedom to move, outdoor access, and the ability to engage in natural behaviors including social interactions, which participants associated with naturalness [40,44]. The majority of participants said that they would choose to buy pork produced in the outdoors as an alternative, based mainly on the perception of higher naturalness and meat quality, as shown in other studies [9,40]. It is well recognized that there are some obstacles for materializing this preference in purchasing behavior, mostly the higher price compared to products from intensive animal production systems [63]. Most participants in this study believed that Brazilian citizens would be willing to pay more for pork from loose pens or outdoor farrowing compared to the current price of pork. Surveys on attitudes and willingness to pay towards livestock production systems are criticized for not directly reflecting food animal products purchase behavior (i.e., the “attitude–behavior gap” [64]). Yet, negative public attitudes towards livestock productions systems and practices may be reflected in citizens’ behaviors, like support for regulation or retailers’ actions [4]. In recent years, the livestock industry has faced competition from alternative animal and non-animal protein sources [65]. This highlights the relevance of promoting naturalness in the animal production systems, given that this aspect of production is highly praised by consumers [66]. Additionally, the perceived naturalness of animal proteins [40,67] and cellular alternatives [68,69] is a key element shaping acceptability and preferences for these different products.

Our sample is balanced and representative of the country population of key demographic variables [29], except education. Although some studies have found that education influences the concern about animal welfare [47,70], the education level of participants had no statistical association with responses in this study. Additionally, this segment may represent opinion holders with substantial purchasing power, traits that may influence changes in production practices. Consistent with the more negative attitude scores of females compared to males in this study, it has been shown that women tend to have higher concern about animal welfare and contentious livestock production practices compared to males [10,47,71]. The changing role of women in society may explain in part the growth in societal concern with animal welfare in both industrialized and emerging countries [4]. As we move towards a more gender-equitable society and the voice of women is increasingly heard, this is an important factor to consider regarding the livestock industry’s license to operate.

We acknowledge the fact that the use of images showing that the loose pens are an indoor housing system may have influenced our results. Studying lay peoples’ assessment of farmed pigs using pictures, Busch et al. [72] showed that picture background had a marked influence on participants’ evaluation of the housing system and pig happiness. Similarly, in a study investigating public attitude toward surgical castration of male piglets and its alternatives participants provided with audiovisual information were more positive to immunocastration, a less aversive castration method compared to surgical castration, than participants who were exposed to only written information [73]. The decision to use images to give participants a context of the systems before asking them to evaluate them was based on previous studies showing that the public has low awareness of pig production systems (e.g., [10,74]), which our findings confirmed. Additionally, this does not deviate from real situations where people are exposed to images of animal production systems used in campaigns aiming to increase public awareness about these systems (e.g., www.endthecageage.eu, accessed on 4 November 2021).

## 5. Conclusions and Recommendations

Participants’ attitudes were most negative towards the farrowing crates and most positive towards outdoor farrowing, and positively correlated with sows’ perceived quality of life. Importantly, avoiding piglet mortality was not viewed as a justification to house sows in crates that restrict movement and other natural behaviors perceived as important elements for sows’ welfare. Loose pens, the main alternative available to replace farrowing crates in confined systems, did not address all concerns shown by participants, including naturalness and the ability of sows to socialize and express all aspects of maternal behavior. For participants, the pig industry stakeholders (farmers, industry, and scientists) have the responsibility to develop housing methods that enable sows’ freedom to move while avoiding piglet mortality by crushing, with many expressing optimism that they will be able to do so. Our findings suggest that public support for loose farrowing housing may be lower than expected, for reasons similar to the enriched cages for laying hens [75,76,77]. However, participants’ trust in the pig industry stakeholders’ capability to develop farrowing systems that are in line with their concerns suggest an opportunity for the pig industry to be proactive in communicating with the public to reach common grounds. Social acceptability of loose housing may be undermined by the use of temporary crating and the retention of some aspects of the original system they aim to replace, namely individual housing and lack of environmental enrichment. Previous researchers [4,28] have warned that making changes in the livestock systems that do not address the issues considered most important by the general public poses an economic risk to farmers and other stakeholders in the supply chain. We conclude that maintaining farrowing crates may erode the pig industry’s social license and suggest that pig industry stakeholders and policymakers need to engage with the public in a two-way communication to ensure that alternative farrowing housing has societal support.

## Figures and Tables

**Figure 1 animals-11-03439-f001:**
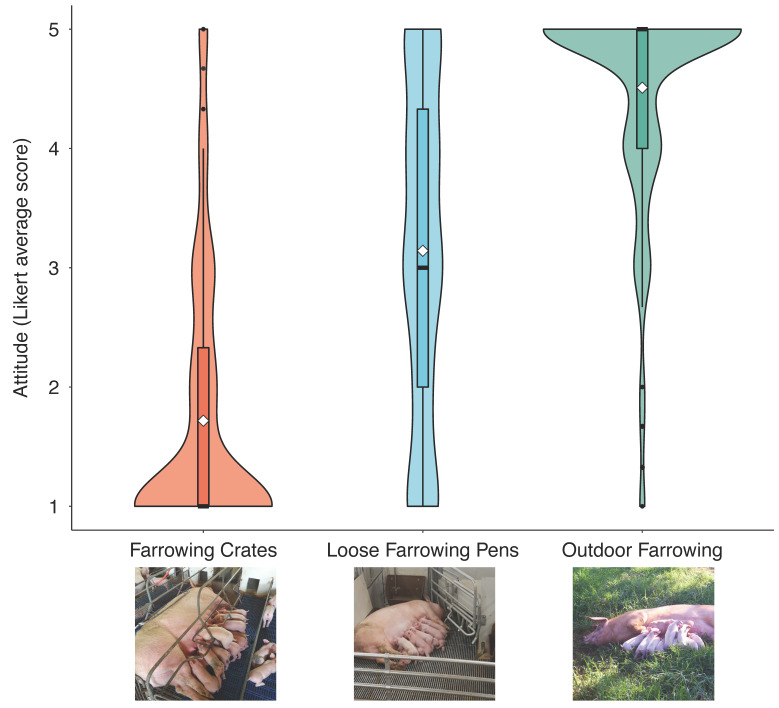
Violin and box plots of participants’ attitude towards three farrowing systems: farrowing crates (*n* = 395), loose farrowing pens (*n* = 384), and outdoor farrowing (*n* = 392). Attitude score is a construct consisting of the average of three 5-point Likert scales, with higher numbers indicating a more positive attitude. The width of the violin represents the density of participants on the respective attitude score. The black dots represent outliers, the white diamonds represent the mean value, and the thickest line in the box plot represents the median value.

**Figure 2 animals-11-03439-f002:**
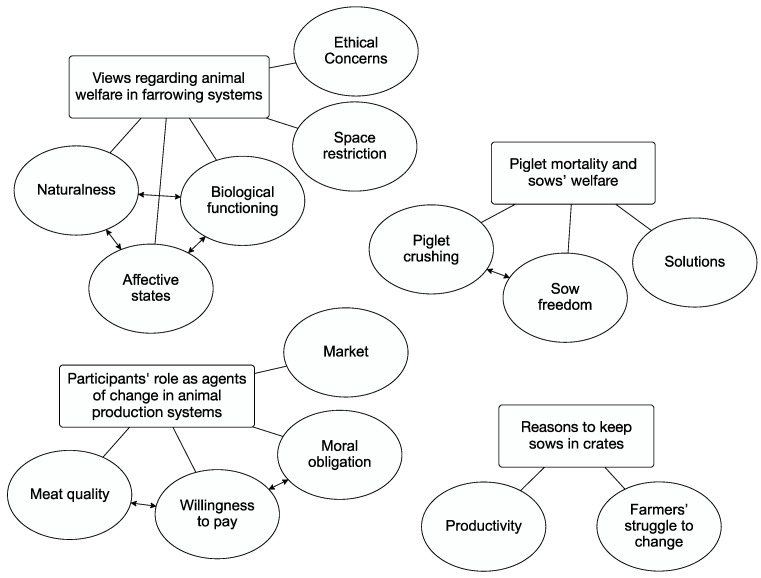
Themes developed based on participants’ open answers. The boxes represent the themes, and the circles represent the underlying codes. The two-way arrows represent relationships between codes.

**Table 1 animals-11-03439-t001:** Demographics of survey participants (*n* = 1171) according to the farrowing system they were assigned to answer about (farrowing crates—FC, loose farrowing pens—LP, and outdoor farrowing—OF) and of the Brazilian population according to latest census (IBGE, [29]).

Variable	FC (*n* = 395)	LP (*n* = 384)	OF (*n* = 392)	Total (*n* = 1171)	IBGE 2010
	(%)	(%)	(%)	(%)	(%)
Sex					
Female	53	48	55	52	51
Male	47	52	45	48	49
Age					
18 to 24 years old	17	21	16	18	16
25 to 34 years old	22	26	24	24	23
35 to 44 years old	28	20	25	25	20
45 to 54 years old	20	17	18	18	18
55 years old and over	13	16	17	15	23
Education					
Up to high school	33	37	35	35	64
Post-secondary education	67	63	65	65	36
Current residence					
Urban	87	91	89	89	85
Rural	13	9	11	11	15
Region of Brazil					
South	24	27	28	27	15
Southeast	49	37	44	43	42
North	2	3	4	3	8
Northeast	17	22	17	19	28
Midwest	8	11	7	8	7
Household income (minimum wage) ^1^					
Up to 2	22	24	26	24	24
2 to 5	29	30	29	29	49
6 to 10	16	12	13	14	14
Over 10	6	9	7	7	13
I prefer not to say	27	25	25	26	

^1^ Data referring to income were taken from the Family Budget Survey 2017–2018 [33].

**Table 2 animals-11-03439-t002:** Factors associated with attitudes towards the farrowing housing systems among participants (*n* = 1171). Attitude score is a construct consisting of the average of three 5-point Likert scales, with higher numbers indicating a more positive attitude.

Factor	Level	*n*	Attitude (Mean)	SE
Housing system	Crates	395	1.64	0.06
	Loose Pens	384	3.02	0.06
	Outdoors	392	4.36	0.07
Sex	Female	611	2.83	0.05
	Male	560	3.19	0.06
Days per week that respondent eats meat	1 to 2 days	211	3.01	0.08
	3 to 4 days	231	3.11	0.07
	5 to 7 days	540	3.24	0.05
	None	40	2.69	0.17
	Rarely	149	2.97	0.09
Previous awareness of the housing system	No	570	2.92	0.06
	Yes	601	3.10	0.06

## Data Availability

The following are available online at https://figshare.com/s/874c68b8e65f6dbd0b5a.

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
