# Peer review of "“Mothers Should Have Freedom of Movement”—Citizens’ Attitudes Regarding Farrowing Housing Systems for Sows and Their Piglets"

_animals, 2021, doi:10.3390/ani11123439_

Round 1

Reviewer 1 Report

Manuscript animals-1473074, entitled “Mothers should have freedom of movement” - Citizens’ attitudes regarding farrowing housing systems for sows and their piglets

Recommendation:       The above paper is not suitable for publication in its present form.

General comment

The article provides useful information about the attitudes of Brazilian citizens regarding farrowing housing systems for sows and their piglets. Although, the study was in general appropriately designed and implemented, there are some points that should be corrected or clarified.

General comment

Please choose normal font or italics for participant views uniformly in text

Minor points:

L35: “Furthermore, loose…”

L46: “imposed” instead of “done”

L51: “…higher level of animal welfare…”

L51: What do you mean by “lead to retailer regulation”?

L54: “banned their” instead of “called for bans on its “

L65: Please check journal reference style

L66: What do you mean by “are still late”?

L89: “…crushing, since it has been shown that neonatal…

L92: “recommended” instead of “proposed”

L93: “…originated from the demand…”

L98: What do you mean by “and underlying reasons”?

L130: “Participants were then invited…”

L133: “…systems is translated to English and is provided…”

L148: “provided with” instead of “given”

L161: “performed” instead of “done”

L164: “…meat consumption preferences: whether…”

L217: “provided” instead of “gave”

L223: Please check reference style

L226-229: The percentages of FC, LP, OF refer to?

L295: Please delete “[33]”

L376: “since” instead of “given that”

L500: “that exist” instead of “there is”

L509-510: Please rephrase

L520: “previously” instead of “by”

L577-579: Please rephrase

L586: How did you reach to the conclusion that education level did not influence attitudes in this study?

L591: “…contentions regarding livestock…”

L610-634: It is not common to use references in this part (L622-623, 628)

L628: “Previous researchers” instead of “Others”

Reviewer 2 Report

Comments on manuscript farrowing crates.

An interesting topic. It is obvious that the constraints of the farrowing crate would be the next focus as many localities continue to phase out the traditional sow stall, putting pressure on the next part of the production system. I feel in some ways that the disconnect between our food production systems and the population lead to some of the misunderstandings that are clearly articulated in this manuscript, in the respondents’ comments. I might also say there is a certain naivety to some of the comments around the ability of science and research to solve problems like neonatal mortality – after all it is both a construct of evolution in a litter bearing species, and a problem that has researched for decades now! The ability of people to understand risk (here the mortality of the piglet) and visualise the consequences is generally poor, and that it picked up in the manuscript – I think that this might allow the industry an opportunity to educate. The parallels with the chicken industry are clear in the differences between cages, enriched cages and free-range. Despite the known disadvantages of free-range systems (and indeed much of those are due to the lack of a clear definition of the density of the system) the argument on enriched cages has been lost a long time ago as far as the public are concerned (as mentioned in the conclusion here).

Of course, the naivety around solutions and risk does not detract from the point of the manuscript – the idea of a social licence to operate is well established, and whether right or wrong this opinions form part of society’s acceptance of these systems.

The methodology seems sound, as does the statistical analysis. In places the results do feel long but that is partly due to the inclusion of the specific supporting comments by the respondents – removal of some of these would reduce the length of the manuscript without losing any of the impact, however that is more a decision for the editor than a reviewer.

Specific comments

Line 70 – “crates to group gestation until 2045” – I think that replacing “until” with “by” makes more sense in this sentence

Line 86 – “believed to be” –Whilst we can argue about why crushing is the ultimate cause of death (under-weight, starvation, hypothermia etc) is it “believed to be” the largest cause of death or is it the most common cause of death?

Line 177 – “attention check” led to some respondents being excluded. This needs some explanation as I cannot see mention of what this is in the manuscript?

Line 240 – table footnote is running into the following text, needs a line inserted between them

Line 356 – Participants views are shown in italics from this point forward, but not beforehand. Consistency.

Line 414> It is mentioned that respondents with a strong anti-farming/meat view point have been excluded from the analysis, and I assume that comments they may have made are also not shown here? I ask as some of the comments are seem rather strong.

Line 596 – good to see acknowledge of the issues that can arise from bias in photographs in studies of this kind. However, in my experience the photograph of the outside farrowing situation seems very idolised. The picture may reflect the standard of Brazilian outdoor piggeries, but pasture does not last long, and most piglets that size would be in an enclosed arc/hut with a sufficiently high door step to keep them in, at least until they get big enough to cope outdoors.  

Line 631 (plus abstract) – the word “essential” is a strong one here. Are there alternative strategies available that could make the discussion around intermediate environments more palatable to the public? “Essential” to my mind means that the social licence will be lost if change is not implemented.  Given the slow progress in removing cage eggs from supermarkets globally, despite pressure from vocal elements of the general population, it suggests that the consumer (or supermarket) drives the system more than the general population. I suspect that as long as the industry promotes the move to more open systems, not necessarily free range/outdoor, the consumer will continue to buy pork products. I am a little cynical of the idea that people saying they will pay more for a product in general, supermarket data might suggest that does not happen in practice?

Reviewer 3 Report

This is an interesting and generally well written account of a questionnaire survey with a large number of respondents who were in most aspects demographically representative of the Brazilian population. The conclusions are compelling and there is no doubt that, on the principle that the customer is always right, majority customer opinion would require pig production systems to adopt practices that allow sows much more freedom of movement and opportunity for natural behaviour.

As the participants in the survey indicated that companies should make changes to more sow-friendly systems, the authors might like to discuss this a little more by referring to two recent articles that provide reassurance on piglet welfare improving in group housing (Kinane et al., Freedom to Grow: Improving SowWelfare also Benefits Piglets, 2021) and sows in group housing having the same level of productivity as individually housed sows (Min et al., Comparison of the Productivity of Primiparous Sows Housed in Individual Stalls and Group Housing Systems, 2021).

One participant correctly identified loss of mothering ability in some modern breeds as a potential problem, which could be addressed, especially for outdoor systems, by using a modern, high-producing outdoor-adapted breed like Duroc that is known for good mothering ability, but that might go a bit beyond the scope of this article.

Line 590/1: ‘contentions’ should be ‘contentious’.
